# Smart Graphene Nanoplatelet Strain Sensor for Natural Frequency Sensing of Stainless Steel (SS304) and Human Health Monitoring

**DOI:** 10.3390/ma15113924

**Published:** 2022-05-31

**Authors:** Debadatta Sethy, Krishnan Balasubramaniam

**Affiliations:** Center for Non-Destructive Evaluation, Indian Institute of Technology Madras, Chennai 600036, India; balas@iitm.ac.in

**Keywords:** graphene nanoplatelets, Lycra textile sensor, spectroscopical characterization, frequency response function, modal shapes

## Abstract

The main purpose of this article is to monitor the natural frequency of stainless steel (SS304) with and without defect by spray-coated smart graphene nanoplatelet (GNPs)-doped poly (methyl methacrylate) (PMMA) nanocomposite strain sensor and human health by smart Lycra denim textile sensor. Methods such as impact hammer test and NI-daq, finite element method (FEM) simulation by Abaqus 6.12, and fast Fourier transform (FFT) study were applied for frequency monitoring of SS304. For human health monitoring, edema disease inspection, cough, and biceps locomotion were studied by graphene sol–gel textile sensor. We report eight sensors fabricated by scotch tape exfoliation method and their sensitivity was checked in terms of gauge factor (GF). The highest and lowest GF-based sensors were checked for sensitivity in the defect (hole) specimen. These sensors were used to sense the natural frequency of SS304 at three different positions in the cantilever beam. The same quantity of GNPs was used for making Lycra denim textile sensors for human health monitoring. The Lycra denim textile sensor showed a 216% change in resistance in the left calf muscle, which is less than right leg flexibility, indicating good sensitivity. In addition, the textile sensor helped in sensing coughing and biceps monitoring. The ease in fabrication and high sensitivity demonstrate the potential ability of GNPs for futuristic smart material for structural and human health monitoring.

## 1. Introduction

Stainless steel of grade SS304 is one of the most versatile widely used stainless steels, having excellent forming and welding characteristics. It is excellent in a range of environments and many corrosive media. SS304 is also resistant to warm chloride environments and to stress corrosion cracking above 60 °C. It also has good weldability by all standard fusion methods both with and without filler metals. Therefore, because of these properties, SS304 has versatile applications in plates and pipelines. It has a lot of applications in beer brewing, milk processing, and winemaking equipment. From kitchen, benches, sinks, troughs equipment, and appliances to architectural paneling, railings, and trim, SS304 is a promising tool for many usages. In thermal power plants, and also mining areas, SS304 is used basically for heat exchangers, woven screens, quarrying, and water filtration, respectively. Even in mechanical devices, such as threaded fasteners and springs, SS304 has contributed a lot. Therefore, its inspection is of high concern for the industry. In advance non-destructive (NDT) techniques such as infrared (IR) thermography, ultrasonic testing, computed tomography (CT), spectroscopy analysis, etc., are used for the inspection of stainless steel. However, these techniques are excellent but not reliable, and have a high maintenance cost. Therefore, to avoid these maintenance costs and poor reliability, a smart sensor made from GNPs is the best solution. Graphene nanoplatelet, being a 2D material, is the derived form of graphite. It reflects the allotrope form of carbon-containing numerous double bonds having an electronic configuration of [He] 2s^2^2p^2^ situated at the p-block in the periodic table. Each atom in a graphene sheet is connected to its three nearest neighbors by a σ-bond and contributes one electron to a conduction band that extends over the whole sheet. This rearrangement of structure and electronic configuration helps in sensing the outer environment. GNP is a crystalline form of the element carbon with its atom arranged in a hexagonal structure. In addition, properties such as high mobility charge carriers, monoatomic thickness, and high surface area help in sensing the health monitoring of structures [1,2,3,4,5]. Graphene has its unique properties due to its electronic band structure which plays a vital role in electronic and optoelectronic devices. This helps in studying the tunable work function. This tunable study has demanded much attention towards graphene usage in the modern era [6]. Graphene has not only limited itself to optoelectronic properties but also can attenuate electromagnetic pollution. It also has good shielding properties because of its structure that helps in transmission loss of the incident electromagnetic wave [7]. Many health monitoring works have been carried out on structures and the human body with graphene [8,9,10,11,12]. Here, in this article, graphene has been suspended over substrate polymethyl methacrylate (PMMA), which is an amorphous and thermoplastic polymer that maintains good hardness and strength in making sensors. In addition, GNPs enhance the microstructure and viscoelastic characteristics of PMMA. Smart sensors, especially made from graphene materials as substrates, have been studied [13,14,15]. Substrates such as PMMA have been studied as reinforcement carbonaceous filler for obtaining well-dispersed alignment-based nanocomposites with GNP. Many graphene-doped PMMA sensors for human health monitoring have been reported from heart rate, pulse oxygenation, blood glucose meter, to electrocardiogram signal, etc., by novel monitoring devices made out of graphene [16]. In addition, thermal management work taking graphene as a sensing indicator has been studied in which PMMA acts as a substrate in GNPs-based nanocomposite sensors [17,18]. In our previous work, the thermal signature was improved by making PMMA a substrate by doping graphene into it. Much structural health monitoring work has been reported by GNP sensors in which excellent GF obtained has been deployed in thermal imaging and also in strain monitoring activity [19,20,21,22]. The graphene-based sensor is also best for piezoresistive sensing which has been proved in many works such as energy harvesting and beam sensing purposes [23,24]. These sensors are brittle but show good response in electromechanical coupling. To overcome the brittleness, many graphene-doped functionalized sensors have been implemented in sensing [25]. Some functionalized materials such as polydimethylsiloxane, and also reduced form of graphene, help not only in high sensitivity but are also capable of proximity sensing capability [26,27,28]. Graphene, being a smart material, has tremendous application not only in piezoelectric response limited to beams but also in textiles, too. Much work has been carried out for finding the natural frequency of graphene platelets but it has not been implemented as a sensor for sensing the natural frequency of structures in terms of piezoresistivity.

Many works have also been carried out, such as silk spandex fabric strain sensor by coating reduced graphene oxide in which performance was measured by performing a cyclic test which remained constant for 1000 cycles [29]. Even some interesting work in gait monitoring has also been reported, where the knee region was monitored by two identical sensors made out of graphene [30]. In the medical field, online data monitoring had been performed on patients for asymmetrical walkway motion, but here it is very costly in sensor installation, which was a photoelectric beam, and also a time cost factor [31]. Many works such as laser-induced graphene have been implemented on the skin for motion monitoring. This is a laser direct writing process whose power consumption and installation cost are too high. Some graphene Ecoflex sandwich structures have been implemented for vocal fold vibration detection purposes where the drop-casting method was used for the production of the sensors [32].

The main objective is to monitor the natural frequency of SS304 in defect (1 mm diameter hole) and without-defect specimens by spray-coating the structure and to monitor human health by sol–gel-coated Lycra denim sensor. Therefore, the article has been divided into three parts; the first one is regarding experiments with an impact hammer for natural frequency calculation with the help of NI-USB daq-4432, and then we study the sensing behavior with GNPs/PMMA sensor. The second is regarding surface modeling to study the modal analysis affecting the resistance behavior during natural frequency sensing. Then, a sol–gel coated graphene sensor is used for studying human body locomotion and edema disease analysis.

## 2. Sensor (GNPs/PMMA) Fabrication

In the fabrication, as shown in Figure 1, of a smart GNPs/PMMA sensor, three chemicals were taken, namely, GNPs of 50 mg (thickness < 2–4 nm; lateral size = 5 µm, received generously from GRAPHENE LAB Ltd., London, UK) were mixed with THF-AR of 100 mL. Then the solution was sonicated for 10 h. Similarly, a solution of PMMA of 2 mg and THF of 20 mL was sonicated for 4 h and then both the solutions were mixed and sonicated for more than 8 h. After sonication, the GNP sensor was ready for fabrication upon SS304. The GNP sensor then was spray-coated and a scotch tape was used for mechanical cleavage to maintain the intrinsic electrical resistance, as shown above in Figure 1f. Therefore, overall, eight GNP spray-coated sensors upon SS304 were fabricated and were mechanically cleavaged to maintain intrinsic resistances such as 200 Ω, 400 Ω, 500 Ω, 650 Ω, 800 Ω, 900 Ω, 1 kΩ, and 3.5 kΩ. These sensors were further tested for sensitivity analysis compared with an industrial strain gauge (HBM1-LY41-6/350 (R_0_ = 0.35 kΩ; measured GF is ~1.6)). 

## 3. Characterization

As shown in the above Figure 2, for characterization, XRD analysis was performed for both pure GNPs, pure PMMA, and GNPs/PMMA/THF. This is the most comprehensive study and characterization to identify unknown materials and is relatively reliable for identifying the structure and composition of any material from the position in terms of degree and intensities of diffraction peak. From the above figure, it is observed that the position of the XRD peak, which correlates to the powder film platelets behavior of GNPs material, falls at the value of 2Θ at 26°. This indicates the presence of GNPs in our raw material, whose microstructure can be visualized from the SEM image as shown in Figure 2b. These GNPs’ (thickness < 2–4 nm; lateral size = 5 µm, received generously from GRAPHENE LAB Ltd., London, UK) properties can be seen in Figure 2c. Similarly, in Figure 2d, XRD was performed also for PMMA and PMMA/GNPs/THF solution. It shows the presence of metha-acrylate, which has three intensity peaks, one at 13.6° another at 23°, and one at 42.6°, which describes the amorphous nature of PMMA. The sensor solution (GNPs/PMMA/THF) also shows the intensity peak because of the presence of PMMA, GNPs, and THF. The narrow peak observed in Figure 2d falling at 25° and 30.6° show the good crystallinity of PMMA. In addition, Figure 2e,f show the morphology of PMMA and GNPs/PMMA. Here, in the PMMA morphology, the crystalline lamellae structure is separated from the amorphous phase, which makes PMMA turn black under SEM [33]. Similarly, GNPs are also doped into PMMA, which can be seen as flakes of sheets on the surface of PMMA. Here, PMMA is acting as the substrate which holds GNPs on it that acts as a smart nanocomposite sensor.

## 4. Experimental Setup

In the experimental setup, as shown in Figure 3, the arrangement was performed for both sensitivity analysis of strain gauge and GNPs/PMMA sensor. These smart sensors were first tested upon Instron-8801 UTM under uniaxial loading at the rate of 1 mm/min at strain-controlled loading, which can be seen in Figure 1j. As shown from Figure 3a, National Instruments-based LabVIEW (version 2016, CNDE Lab, IIT-Madras, Chennai, India) software is interfaced with NI-DAQ USB-4432 for data collection at the sampling frequency of 1000, and correspondingly, a Keithley SourceMeter-2450 was used for resistance data collection. In Figure 3b,c, an impact hammer (DYTRAN_DYNAPULSE, Chatsworth, CA, USA) is shown and it is attached with a BNC coaxial connector to USB-4432 at one end and the accelerometer is connected to the other end of the USB-4432. Figure 3d,e shows the smart sensor from GNPs/PMMA and industrial strain gauge fabricated upon SS304, which senses the change in resistance during natural frequency excitation. 

## 5. Results and Discussion

### 5.1. Natural Frequency Sensing of SS304

As shown in the above Figure 4a,b, first of all, eight specimens were spray-coated with the GNPs/PMMA-based sensor. Then, these sensors were exfoliated mechanically by scotch tape, as already explained in Figure 1. Then, these sensors were brought under intrinsic resistances of 200 Ω, 400 Ω, 500 Ω, 650 Ω, 800 Ω, 900 Ω, 1 kΩ, and 3.5 kΩ, and these sensors were checked then for sensitivity analysis against an industrial strain gauge (350 Ω). 

Here, sensitivity is checked under piezoresistivity action in terms of gauge factor (GF). GF is a parameter that correlates electrical resistance with the mechanical strain which is expressed always as
（1）$$\mathrm{Sensitivity}\;(\mathrm{GF})=\frac{\frac{\Delta R}{R}\%}{\frac{\Delta L}{L}\%}$$

Here, $\frac{\Delta R}{R}\%$ is the normalized resistance and $\frac{\Delta L}{L}\%$ is the normalized strain. These embedded sensors upon SS304 were tested at UTM Instron-8801 under monotonic load conditions at a crosshead displacement rate of 1 mm/min within the elastic range of SS304. 

As observed from Figure 4a,b, the GF for 200 Ω, 400 Ω, 500 Ω, 650 Ω, 800 Ω, 900 Ω, 1 kΩ, and 3.5 kΩ are 52, 263, 36, 141, 84, 116, 65, and 25, respectively. The maximum strain achieved during uniaxial loading within elastic range was 0.15% for 18 s and load of 9500N. These GNPs/PMMA sensors have higher sensitivity than that of industrial strain gauge of 350 Ω, which is 1.5. Now, as the strain increases, the electrical resistance also increases, and hence GF becomes high. As shown in Figure 4c,d, the highest and lowest GF obtained from Figure 4a underwent sensitivity analysis for defect (1 mm diameter hole) based on SS304. The GF obtained from uniaxial loading is 86, 16, and 0.64 for 3.5 kΩ, 400 Ω, and industrial strain gauge, respectively. These higher and lower GF are further used for sensing impact hammer testing during natural frequency finding of a SS304 strip, as shown in Figure 5. 

As shown in Figure 5a, the SS304 strip is fixed to one end by a clamp, and 3.5 kΩ, 400 Ω, and strain gauge (350 Ω) are fixed at a distance of 15 mm from the mid-center of the specimen on the hole (defect) and without-defect specimen. These are used on the specimen to find peak amplitude (resistance) at the respective position of impact hammering. In the experiment, impact hammer (DYTRAN) and shear accelerometer are used for impulse and response behavior with respect to time by interfacing NI-DAQ LabVIEW 2016 for obtaining data at the sampling frequency of 1000. Then, the impact hammer was tapped (perturbed) upon both the defect and without-defect specimen, and the Keithley SourceMeter-2450 (Tektronix, Beaverton, OR, USA) was used for resistance data collection. Then, force data and acceleration data were used for modal frequency calculation by taking FFT in terms of frequency response function (FRF) from Matlab 2016. After processing the data, natural frequencies were found in terms of resistance change, as shown in Figure 5c–h. It is observed from Figure 5, from without-defect specimen, that the sensing of the amplitude of resistance peak varies with varying of deflection during tapping with an impact hammer. For strain gauge (350 Ω), the highest peak of resistance amplitude was found to be at the free end at position 1, as shown in Figure 5a. At positions 1, 2, and 3, the peak resistance changes of the strain gauge rose to 0.115 Ω, 0.02844 Ω, and 0.007904 Ω, respectively, for without defect, as observed in Figure 5c. Similarly, for the defect sample, as seen from Figure 5f, the peak resistance dropped down to 0.02713 Ω, −0.0278 Ω, and −0.06711 Ω, respectively, for positions 1, 2, and 3. Therefore, the % decrease in resistance of the defect specimen from without-defect specimen fell to 76.4%, 197.74%, and 949.06%, respectively, whereas when sensed with GNPs/PMMA sensor of 400 Ω, it was found that for the without-defect specimen, as seen from Figure 5d, at positions 1, 2, and 3, the peak rose to 9.927 Ω, 8.859 Ω, and 6.738 Ω. For the defect specimen, as observed from Figure 5g, the peak rose to 9.761 Ω, 5.832 Ω, and 5.455 Ω. Therefore, the % decrease of resistance from without defect to defect fell to 1.67%, 34.16%, and 19.04%, respectively, corresponding to positions 1, 2, and 3. Similarly, for 3.5 kΩ, as observed in Figure 5e,h, the resistance decreased down to 11.23 Ω, 6.672 Ω, and 3.343 Ω from 54.81 Ω, 51.16 Ω, and 22.26 Ω with the % decrement of 79.511%, 89.9586%, and 84.982%, respectively. This % decrement of resistance can be well observed from the histogram shown in Figure 5b and quantitative data analyzed in Table 1.

To understand the interface mechanism, a simulation study was carried out on Abaqus software 6.12 for finding the natural frequency. The natural frequency (f_n_) is obtained from the Euler–Bernoulli equation of cantilever fixed-free thin beam [34,35], generally defined as the below equation:


(2)
$$\frac{\partial^{4}y}{\partial x^{4}}+(\frac{\rho A}{\mathrm{EI}})\;\frac{\partial^{2}y}{\partial t^{2}}=0$$


From the above fourth-order equation, mass (m) = *ρA* (density × area) and the natural frequency (f_n_) are defined as the frequency at which a system tends to oscillate in the absence of any driving or damping force, and is expressed mathematically as
(3)$$f_{n}=\frac{{C_{i}}^{2}}{2\pi }\sqrt{\frac{\mathrm{EI}}{mL^{4}}}$$
where E is the Young’s modulus of the SS304 strip, I is the moment of inertia, m is the mass of SS304, L is the length of specimen, and *C_i_* is the mode of vibration. The values of the mode of vibration are expressed below:
(4)$$C_{i}=\beta_{i}L=1.875,4.694,7.85,11\ldots \;\ldots \;\ldots \;\ldots \mathrm{etc}.$$


Correspondingly, FRF was analyzed without-defect specimen and with defect (hole) specimen after finding FFT from acceleration vs. time data. As shown in Figure 6b, at position 1 (free end), the impact hammering was performed with a force of 20.32N with an acceleration of 14.6 m/s^2^ and, finally, it was reduced down logarithmically. In addition, for the frequency values obtained, as seen from Figure 6a, the modest frequency peak was found to be in the region 21.4 Hz following the next mode with a magnitude of 33.8 Hz. The third mode was found to be 157.2 Hz and the fourth mode was 189.8 Hz. The fifth mode was found to be 258 Hz and the sixth mode was 301.8 Hz. The force amplitudes for corresponding modes are found to be 0.02703N, 0.0333N, 0.02088N, 0.04014N, 0.0452N, and 0.4137N, after performing FFT. Similarly, for position 2, the impact hammering was performed with 8.141 N force and acceleration of 20.92 m/s^2^, as shown in Figure 6d. The first mode was found to be at 28.4 Hz following the second mode at 126.8 Hz. The third, fourth, fifth, and sixth were found to be 160.4 Hz, 175.6 Hz, 191 Hz, and 210.6 Hz, respectively. The corresponding force amplitudes were 0.0371N, 0.044N, 0.05667N, 0.0329N, 0.0328N, and 0.05519N. In addition, third hammering was performed at position 3 near the fixed end. As seen in Figure 6e,f, the specimen was excited with a force at 8.559N and acceleration of 2.486 m/s^2^. The natural frequencies for modes first, second, third, fourth, fifth, and sixth are 29.8 Hz, 50.4 Hz, 67.2 Hz, 166 Hz, 189.2 Hz, and 217.8 Hz. The corresponding force values for the modes are 0.01303N, 0.03925N, 0.06108N, 0.03337N, 0.0353N, and 0.05745N. Hence, it is observed that the natural frequencies value increases with an increase in force value. Similarly, the experiment was performed for defect (hole) specimens. 

As observed in Figure 7a,b, the maximum force is reached at 13.46N and slowly reduced down to −7.989N. Then, the force is reduced down to 6.543N to −1433N in the next decrement cycle, and slowly down to 2.181 N in a decrement manner. Therefore, the acceleration is also retarded slowly from 14.18 m/s^2^ to −33.58 m/s^2^, and consecutively the acceleration is retarded further down to 6.178 m/s^2^. Then it came down further to 3.794 m/s^2^ and then to 3.103 m/s^2^, and came near to rest after logarithmic decrement in the further displacement down to 0.84 m/s^2^. 

In the FFT study between the acceleration and frequency, the frequency was found from the FRF graph obtained as shown in Figure 7a,b. The modest peak amplitude is observed at a frequency of 24.6 Hz, the next peak is observed at 32.8 Hz, and the third peak is at 84 Hz. Similarly, frequencies at higher-order modes were observed at 109 Hz, 133.2 Hz, and 157.6 Hz for fourth, fifth, and sixth respectively. As seen from the force vs. frequency graph, it is observed that the corresponding frequencies increase with increment of force amplitude at 0.023N, 0.01781N, 0.06472N, 0.0605N, 0.1745N, and 0.053N. Similarly, for position 2, as seen in Figure 7c,d, the force is reduced from 5.076N to 7.081N and correspondingly the acceleration is retarded down from 14.94 m/s^2^ to 3.176 m/s^2^, and slowly the retardation of the cycle reached 0.292 m/s^2^. The corresponding frequency obtained from modes of vibration at first mode is at 33.4 Hz, following the second order frequency at 50.4 Hz. The corresponding frequencies at higher order modes are 67.2 Hz, 146 Hz, 173 Hz, and 184.2 Hz. The modest peak is observed at a frequency of 146 Hz and 173 Hz during the time of impact hammering. When the corresponding FFT data between force and frequency are plotted, it is found that the force function is increased vs. frequencies. The force amplitudes are 0.1173N, 0.05745 N, 0.09897N, 0.09117N, 0.02483N, and 0.04178N. Similarly, for position 3, as seen in Figure 7e,f, the force and acceleration values are 16.97N and 1.825 m/s^2^, respectively. After performing FFT of acceleration and frequency, the frequency values obtained are 33.6 Hz, 49 Hz, 86.8 Hz, 132.8 Hz, 209 Hz, and 250.6 Hz, respectively, for first, second, third, fourth, fifth, and sixth modes. Similarly, the force amplitudes also increase with the increment of frequencies. For the first, second, third, fourth, fifth, and sixth modes of vibration, the corresponding force amplitudes after FFT are 0.0576N, 0.05195N, 0.06883N, 0.06543N, 0.0879N, and 0.09497N. 

As seen in Figure 8a,b, mode shapes have been shown. Here, 2D and 3D mode shapes are shown for specimens without defects. Here, the mode shapes are of different shapes, as can be seen from simulation modeling by Abaqus 6.12. In mode 1, the shape is similar to an extension and for mode 2, skin surface went down, making a compressive pattern in mode shapes. Hence, piezo action comes into play and the GNPs/PMMA sensor gives output in terms of change in resistance, as already explained in Figure 5. Similarly, in mode 3, mode 4, mode 5 and mode 6, frequencies are higher as compared to mode 1 and mode 2. The graphene-based sensor is highly sensitive to the beam’s bending action. Therefore, the modes in the beam’s vibration lead to a change in resistance. These modes make the GNPs particle disoriented, and resistance changes from the initial value to a higher range with modes of vibration. Similarly, in the case of the specimen with a defect (hole), the sensitivity of the GNPs/PMMA sensor is reduced to a lower % decrease in resistance, as explained above in Figure 5. The % increase of resistance is more for the without-defect specimen as compared to the defect (hole) one. 

### 5.2. Human Health Monitoring

Here, GNPs are coated with Lycra denim textiles by sol–gel solutions. First of all, Lycra denim textile of ASTM D-5035 was cut into pieces of dimension 150 mm × 30 mm. Then, the Lycra denim textile was rinsed with running water for 5 min. Then it is dried and after drying it was purified again with ethanol by dipping it for 1 h. This completes the cleansing action by anhydrous ethanol. Then, graphene nanoplatelets of 100 mg weight were added to a beaker and 300 mg of sodium lauryl sulfate (SDS) was added. Then, the beaker was set for 3 h under ultrasonic homogenizer at 200 W with 30% amplitude at a frequency of 50 Hz. This helps in the dispersion of GNPs with SDS. Hence, this completes the action of a dispersing agent. Then, in an Erlenmeyer flask, a separate solution for organo-silicon sol was prepared. This is the process for reagent preparation in which 37.28 gm of diethoxydimethyl silane (DEDMS) was poured into 53 mL of 2% aq. solution of polyvinyl alcohol (PVA). Then, 0.28 mL of aluminum isopropoxide mixed with isopropanol (0.015 mol/L) was added and the solution was kept for 2 h at room temperature. After 2 h, the reagent solution was then mixed with 0.8 gm of tetraethoxysilane (TEOS) and 4.1 gm of 3-glycidoxypropyl triethoxysilane (GPS) was added to it. Then the whole solution was mixed with dispersing agent solution. 

Then, the solution was set for 8 h under a magnetic stirrer. Then, after 8 h, the Lycra denim textile cloth was dipped into it and was left for 1 h to be coated completely. Then, the cloth was taken out and was washed under running water. Then, the sol–gel coated textile was dried, and hence the smart sensor Lycra denim textile was ready for experimentation. This can be well observed in Figure 9a–c. Here, basically the smart textile’s main objective is to develop a wearable textile that can sense the movement of the biceps, throat, and calf muscles of the leg in the human body. The biceps brachii muscle is one of the chief muscles of the arm whose origin is at the scapula and insertion into the radius of the biceps. Therefore, this itself comes into play between the shoulder joint and the elbow joint. This locomotion in the forearm make the bicep between joints perform flexion and supination action that brings tensile and compressive action in the mechanical sense [36,37,38]. Therefore, this compressive action and tensile action in the forearm makes the biceps develop piezoaction on the surface of the Lycra denim smart textile sensor. The output is obtained in terms of piezoresistive signals that are collected with the Keithley SourceMeter. The spandex-based wrapped cotton Lycra denim cloth was prepared according to ASTM-D5035 standard having a length of 150 mm, a width of 30 mm, and the gauge length of 95 mm, as shown in Figure 10f, and then lead wires were attached with silver paste, as can be seen from Figure 10g. As seen in the above Figure 10a–c, SEM images were taken for sol–gel-coated smart textile at 500 μm, 500 μm, and 100 μm, respectively. The sol–gel GNPs-based Lycra denim textile sensor senses in terms of change in electrical resistance during piezoresistive action. The smart textile sensor, having an intrinsic resistance of 23 MΩ after using the same wt% of GNPs that was 50 mg, was used for textile sensor fabrication. As seen in Figure 10d,e, EDAX was performed from SEM analysis to know the carbon (C%) and oxygen (O%) after sol–gel coating which were found to be 47.9% and 52.15%, respectively. Then, the textile sensor was tested for sensitivity analysis, as shown in Figure 10h,i. The smart textile sensor underwent sensitivity analysis by Instron (8801) at the displacement rate of 1 mm/min, 2 mm/min, and 3 mm/min, as shown in Figure 10l–n. The GF, as mentioned in Equation (1), for the Lycra textile sensor, was found to be 1.5, which had repeatability at 2 mm/min and 3 mm/min strain testing. This can be well observed in Figure 10k. First of all, three Lycra denim textiles of ASTM-D5035 were cut and tested for tensile strength till the break at a different displacement rate of 1 mm/min, 2 mm/min, and 3 mm/min. At 1 mm/min, the load reached 8.13N at 1136s up to the elastic limit, and then after crossing the plastic deformation, reached breakpoint load at 398N at 2860 s, and then the cloth was torn due to elastane breakage in between the fibers. Similarly, at 2 mm/min of displacement rate, elastic limit reached 4N at 398 s, which then, after crossing plastic deformation, reached breakpoint load of 310N at 1123 s and then elastane fibers were ruptured, and for the displacement rate of 3 mm/min, the load at the elastic point is 6.68N at 230 s which, later after plastic deformation, reached breakpoint load of 347.8N at 715 s. 

Correspondingly, the tensile extension up to an elastic point for 1 mm/min was 8.606 mm at 8.606 s. Similarly, for 2 mm/min and 3 mm/min, the tensile extension was 13.61 mm at 13.61 s and 17.98 mm at 17.98 s, respectively. In each case of strain testing, the tensile strength was found to be the same, that is, 10.245 MPa with ultimate tensile strength (UTS), as 13.27 MPa, as shown in Figure 10i. Therefore, the sensitivity analysis was performed in the linear zone all within the range of the elastic zone. Here, GF was found to be the same as 2.5 in the linear region slope of tensile extension with 8.606 mm, 13.6 mm, and 17.98 mm for 1 mm/min, 2 mm/min, and 3 mm/min, respectively, as shown in Figure 10k. During tensile extension of Lycra denim textile, the elastane fibers inside the cotton wrap stretch and lose their elasticity once it crosses the elastic limit, and plastic deformation leads to tensile failure of the Lycra textile fabric, as shown in Figure 10l–n. Here, at spots 1, 2, and 3, as seen, the fabrics are ruptured and gaps can be seen, signifying the failure of elasticity. Therefore, the sensing of Lycra denim should be achieved within its elasticity limit. This smart Lycra denim textile sensor in this article has been explained for sensing physiological locomotion in the biceps, neck, and leg calf muscle in edema conditions. 

Edema is a disease that is water retention being trapped under muscles in the human body, which can be seen in Figure 11c,d. It is a type of swelling caused by excessive fluid trapped in our body’s tissues. It is the bridge between capillary filtration (lymph formation) and lymphatic drainage. If lymph formation exceeds lymphatic drainage due to either increment of capillary filtration and non-uniformity of lymphatic flow, it leads to raising of venous pressure [39,40]. This, later on, causes venous obstruction as a clot. The consequences of edema lead to calf pain and difficulty in walking. The locomotion in the calf muscle is governed by three parameters, namely, muscle, tendons, and joints. These all together lead to the shortening and lengthening of muscle fibers. Hence, the concentric and eccentric mechanism starts from regular standing to standing on tiptoes and lowering heels slowly back to the floor from tiptoes, respectively. The gastrocnemius and soleus muscles taper and merge at the base of the calf muscle called the Achilles tendon. This tendon helps in stretching the calf muscle, and on relaxing, tends to bring back the muscle to its original position. This repetition of tensile-compressive nature in the calf muscle leads to fatigue cycle, which is sensed by GNPs sol–gel smart-coated textile. Here, the sensor has been employed at the top of the edema region. 

The flexibility of both the legs was checked with electrical resistance data collected from the Lycra sensor with the Keithley SourceMeter during piezoresistance action, as shown in Figure 11a. This graph shows the sensitivity of the edema condition in terms of piezoresistive action. The cycle in the graph is tensile–tensile, in which 34 cycles were played in stretching each leg and bringing it down to its original position for 400 s. The normalized resistance change in baseline shifting was followed by 145.5% during cycling loading in leg locomotion from both the legs. Here, the peak rise of amplitude in terms of normalized resistance is very low in the case of the left leg (361.5% from baseline shifting), but for the right leg, the peak rise in normalized resistance is 520.6%. This delayed rise in the peak of normalized resistance monitors the structural health of the calf muscle, which again was verified with Doppler ultrasonography, as shown in Figure 11g–i. It is an imaging technique that uses sound waves to show blood moving through blood vessels. Therefore, it works by measuring sound waves that are reflected from moving objects such as red blood cells. This is known as the Doppler effect [41]. In Figure 11g, there is a big lumped circular black spot mass in the standing condition which shows blockage in blood flow which is absent in the right leg. As shown in Figure 11h, the ankle also shows a lot of spots blocking the blood flow path. The arteries of the left lower limb show a little swollen appearance of the artery walls. Minimal subcantaneous edema is seen from the left leg and foot, but the appearance is mildly hyperechoic compared to the right calf region. Therefore, the superficial venous system appears mildly prominent in the left lower limb. Few incompetent perforators are seen in the medial and posterior aspects of the leg. Therefore, there is a possibility of mild inflammatory changes. This indicates that image analysis matches with normalized resistance that leads to confirmation of our GNPs-based Lycra textile sensor in sensing the edema. As shown in Figure 11b, for the bicep, the change in resistance amplitude rose to 1.64 × 10^5^ Ω from the baseline shifting of 2.499 × 10^4^ Ω. For 20 cycles of testing in the biceps, the piezoresistive action between the sensor and bicep muscle leads to a change in resistance in cyclic loading in the biceps. Similarly, the sensitivity was checked for the throat region during coughing, in which the change in resistance shifted from a baseline of 2.932 × 10^5^ Ω to 4.109 × 10^5^ Ω, and it was repeated for two cycles. Here, coughing condition was not uniform because of irregularity in larynx motion and the surface of the textile sensor. For the third cyclic behavior at 54 s, it tended to rise at the same amplitude as first cyclic loading, that is, to 4.109 × 10^5^ Ω, but it fell to 2.016 × 10^5^ Ω due to irregularity in locomotion of the larynx. It rose again after 60 s, and again it rose to 3.651 × 10^5^ Ω, and the cycle of coughing ended after falling from 3.54 × 10^5^ Ω to 3.092 × 10^5^ Ω. This is shown in Figure 11e,f. 

## 6. Conclusions

Therefore, it is concluded that the sensitivity in the case of the GNPs-doped PMMA strain sensor during natural frequency varies with impact hammering at different positions. We observed that the % decrease in resistance for the GNPs/PMMA sensor of 3.5 kΩ and 400 Ω is relatively low, which is the relative difference between the defect and without-defect specimen, but in case of industrial strain gauge, the % decrease in resistance is more, compared to 3.5 kΩ and 400 Ω. In the case of smart denim textile sensors, the sensing ability of sol–gel-coated GNP textile with the same wt% of GNPs helps in sensing edema, neck, and bicep locomotion. The locomotion of left and right calf muscles were sensed for edema disease confirmation. In this, the left leg locomotion showed less peak in amplitude change than the right calf muscle. The left leg locomotion was monitored with 216% amplitude normalized, whereas in the right leg, it was 375.1% amplitude. The left leg had less locomotion, which was verified with Doppler ultrasonography, and in the neck, the peak rising and falling was detected with fall of amplitude as 4.109 × 10^5^ Ω, 3.651 × 10^5^ Ω, and 3.54 × 10^5^ Ω gradually, with a decrement of larynx movement. In addition, the bicep locomotion was also monitored successfully in every tensile stretching of muscle fibers. Therefore, it is experimentally proven that the GNP-based sensor acts as a promising indicator for health monitoring not only of SS304, but also of human health monitoring.

## Figures and Tables

**Figure 1 materials-15-03924-f001:**
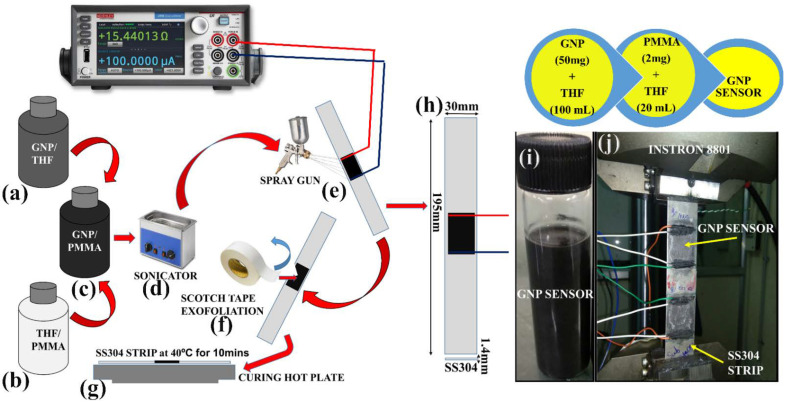
Experimental setup for smart GNPs/PMMA sensor: (**a**–**g**) Sensor preparation from ultra-sonication to spray-coating approach upon SS304 specimen; (**h**) SS304 specimen with geometrical dimensions (195 mm × 30 mm × 1.143 mm); (**i**,**j**) GNPs/PMMA spray-coated sensor for sensitivity test upon SS304 by UTM INSTRON (8801).

**Figure 2 materials-15-03924-f002:**
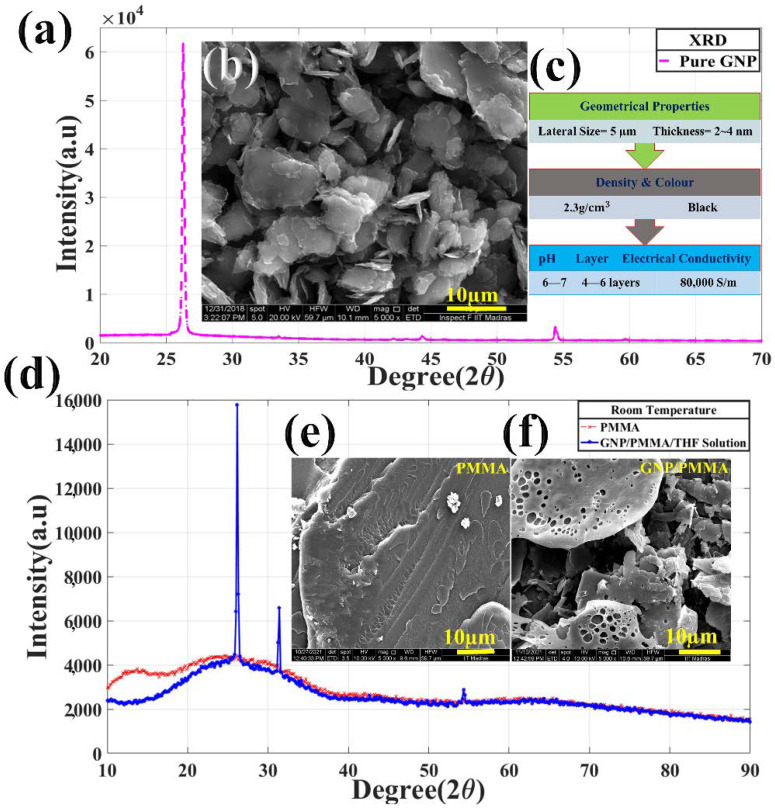
Experimental setup for thermal monitoring during tensile loading: (**a**) XRD of pure GNPs; (**b**) SEM image of GNPs; (**c**) properties of GNPs bought from (GRAPHENE LAB Ltd., London, UK); (**d**) XRD of pure PMMA and GNPs/PMMA/THF solution; (**e**,**f**) SEM image of PMMA and PMMA-doped GNPs.

**Figure 3 materials-15-03924-f003:**
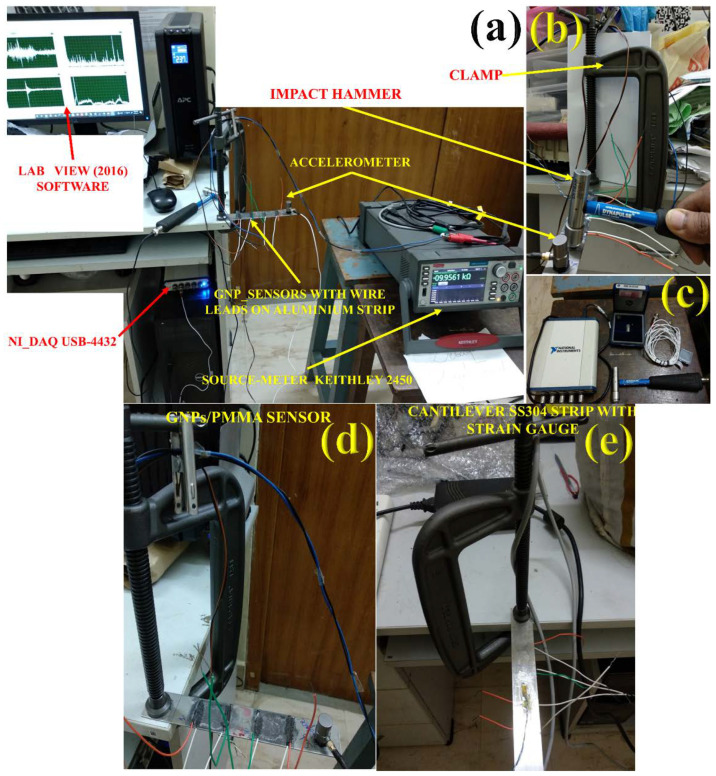
Experimental setup for natural frequency calculation of SS304 strip: (**a**) LabVIEW (2016) software and NI-DAQ USB-4432 used for natural frequency calculation with Keithley SourceMeter-2450 for electrical resistance calculation; (**b**) clamp being used for fixing SS304 strip with accelerometer and impact hammer; (**c**) electronic gadgets for impact hammer method (DAQ USB-4432), wax for fixing accelerometer, cable chords for accelerometer and impact hammer; (**d**) GNPs/PMMA sensor fabricated and accelerometer fixed upon SS304; (**e**) industrial strain gauge fixed to SS304 for comparative study.

**Figure 4 materials-15-03924-f004:**
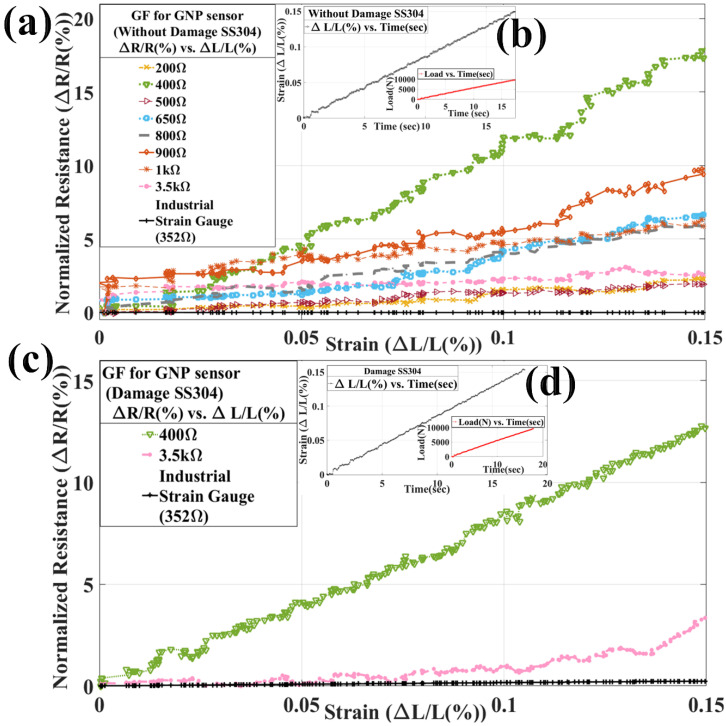
Graph showing gauge factor (GF) with defect and without-defect SS304 specimen: (**a**,**b**) GF calculation by GNPs/PMMA sensor without-defect specimen SS304 and corresponding strain data with respect to time, respectively; (**c**,**d**) GF calculation from highest and lowest GF GNPs/PMMA sensor with defect (hole) specimen SS304 and corresponding strain data with respect to time.

**Figure 5 materials-15-03924-f005:**
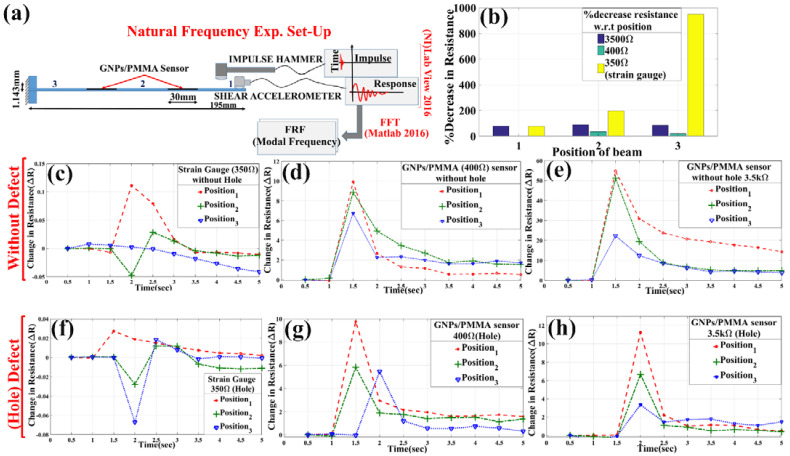
Experimental setup for finding natural frequency: (**a**) Impulse hammer and shear accelerometer used for natural frequency finding with attached GNPs/PMMA sensor; (**b**) histogram showing % decrease of resistance from hole specimen by sensors; (**c**–**e**) change in electrical resistance by industrial strain gauge, 400 Ω and 3.5 kΩ GNPs/PMMA sensor at without-defect SS304; (**f**–**h**) change in electrical resistance by industrial strain gauge, 400 Ω and 3.5 kΩ GNPs/PMMA sensor at with (hole) defect SS304.

**Figure 6 materials-15-03924-f006:**
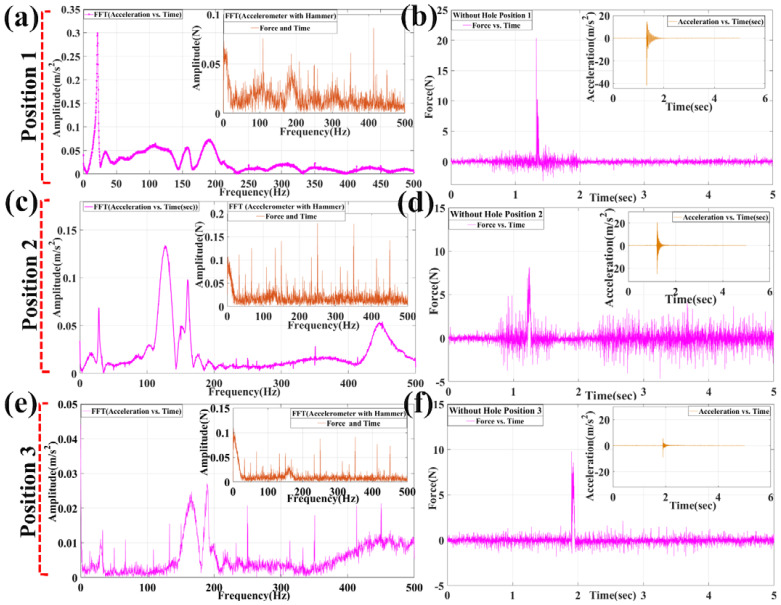
FFT graph plotted by MATLAB 2016 for obtaining modes of frequencies from acceleration and force amplitude with respect to time and its corresponding magnitude graphs at different positions of impact hammering (without hole): (**a**,**b**) FFT graphs and magnitude graphs for acceleration and force for position 1; (**c**,**d**) FFT graphs and magnitude graphs for acceleration and force for position 2; (**e**,**f**) FFT graphs and magnitude graphs for acceleration and force for position 3.

**Figure 7 materials-15-03924-f007:**
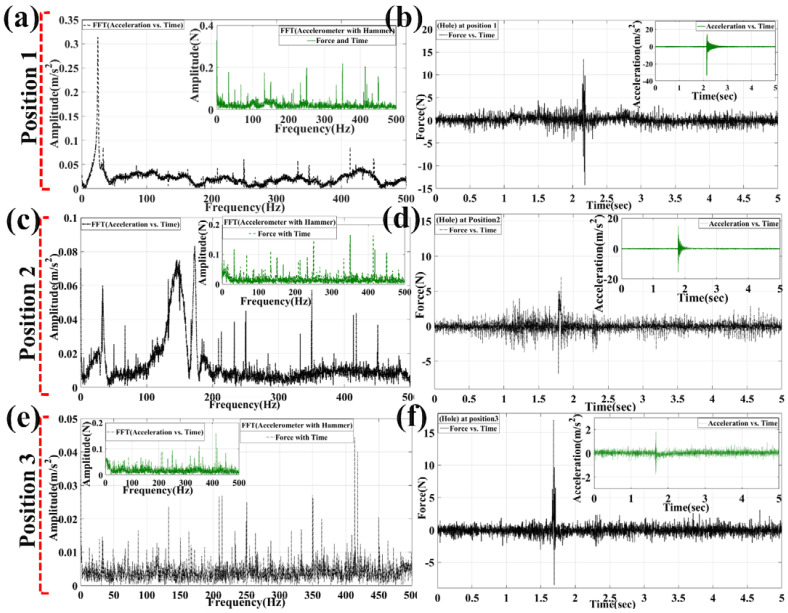
FFT graph plotted by MATLAB 2016 for obtaining modes of frequencies from acceleration and force amplitude with respect to time and its corresponding magnitude graphs at different positions of impact hammering (with (hole) defect): (**a**,**b**) FFT graphs and magnitude graphs for acceleration and force for position 1; (**c**,**d**) FFT graphs and magnitude graphs for acceleration and force for position 2; (**e**,**f**) FFT graphs and magnitude graphs for acceleration and force for position 3.

**Figure 8 materials-15-03924-f008:**
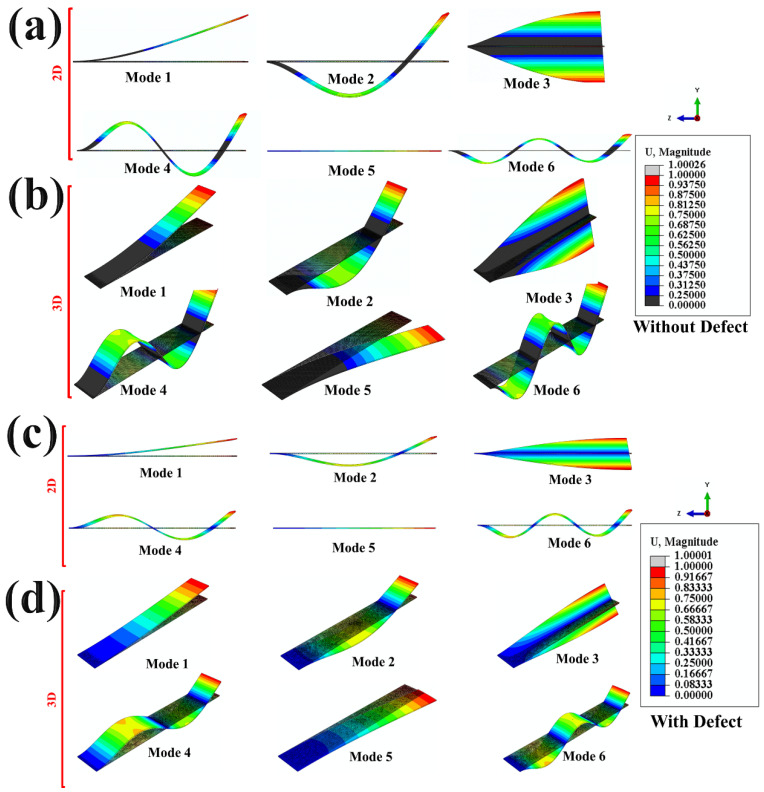
Modal shapes obtained through FEM (Abaqus 6.12) simulation. (**a**,**b**) 2D and 3D mode shapes of all frequencies for without defect (hole); (**c**,**d**) 2D and 3D mode shapes of all frequencies with defect (hole).

**Figure 9 materials-15-03924-f009:**
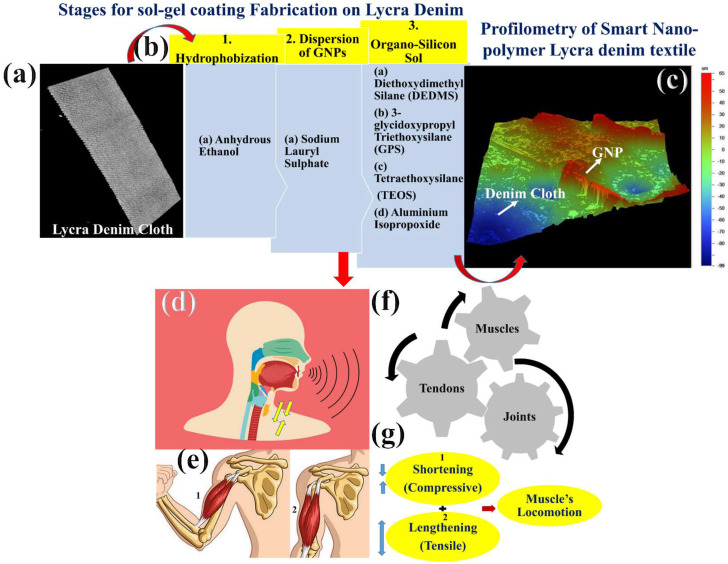
Fabrication of smart Lycra denim textile cloth and profilometry study: (**a**,**b**) computed tomography of Lycra denim cloth and b chemicals used for sol–gel coating fabrication upon Lycra denim textile; (**c**) profilometry used for measurement of thickness of coating; (**d**,**e**) sensor location for wrapping around neck and bicep zone; (**f**,**g**) schematic for human body locomotion with muscles, tendons, and joints during compressive and tensile action.

**Figure 10 materials-15-03924-f010:**
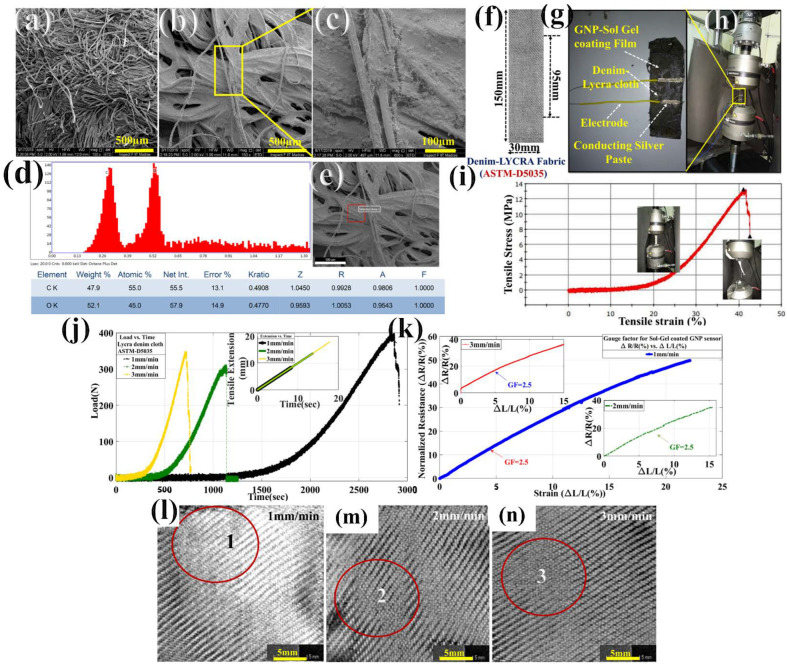
SEM study and sensitivity analysis of smart sol–gel-coated Lycra denim textile sensor: (**a**–**c**) SEM image taken at 500 μm and 100 μm; (**d**,**e**) EDAX for elemental mapping; (**f**–**i**) ASTMD5035-based Lycra denim textile for sensitivity testing under INSTRON-8801; (**j**) load and extension data with respect to time; (**k**) sensitivity test at different displacement rate; (**l**–**n**) CT image testing at different displacement rates.

**Figure 11 materials-15-03924-f011:**
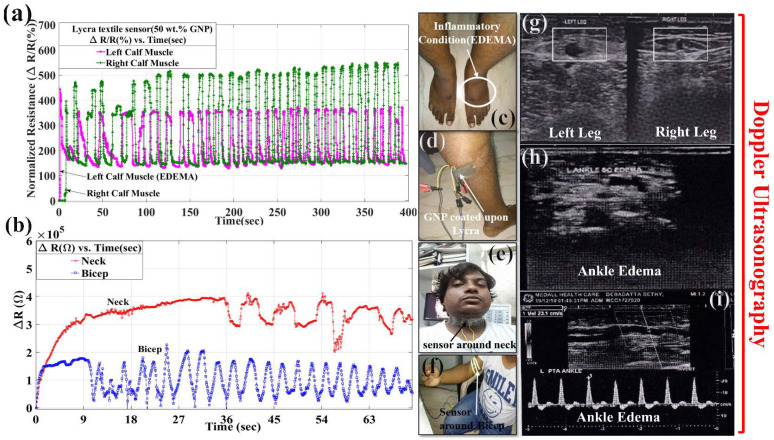
Experimental setup for electrical resistance data analysis by wearable Lycra denim sensor coated with GNPs and sol–gel solutions: (**a**) Normalized resistance change at left and right calf muscles for edema detection by smart Lycra textile sensor (coated with GNPs and sol–gel solutions); (**b**) change in resistance for neck and biceps detected by smart Lycra textile sensor (coated with GNPs and sol–gel solutions); (**c**–**f**) inflammation showing edema at ankle**,** Lycra textile sensor attached at calf muscle, neck, and biceps, respectively; (**g**–**i**) Doppler ultrasonography for confirmation of EDEMA.

**Table 1 materials-15-03924-t001:** Quantitative analysis of electrical resistance during impact hammering at different positions.

Gauge Factor with Resistance	Position of Impact Hammering (Natural Frequency)	Without Defect (ΔR(Ω))	With Defect (Hole) ΔR(Ω))	% Decrease in Resistance Change (ΔR%)	Difference in Resistance Change (ΔR(Ω))
3.5 kΩ	1	54.81	11.23	79.511	−43.58
	2	51.16	6.672	89.9586	−44.488
	3	22.26	3.343	84.982	−18.917
400 Ω	1	9.927	9.761	1.672207	−0.166
	2	8.859	5.832	34.16	−3.027
	3	6.738	5.455	19.04126	−1.283
350 Ω_Industrial strain gauge	1	0.115	0.02713	76.40	−0.08787
	2	0.02844	−0.0278	197.7496	−0.05624
	3	0.007904	−0.06711	949.064	−0.075014

## Data Availability

Not applicable.

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
