# Peer review of "Smart Graphene Nanoplatelet Strain Sensor for Natural Frequency Sensing of Stainless Steel (SS304) and Human Health Monitoring"

_materials, 2022, doi:10.3390/ma15113924_

Round 1

Reviewer 1 Report

The work of Sethy et al. titled “Smart graphene nanoplatelet strain sensor for natural frequency sensing of stainless steel (SS304) and human health monitoring” is detailed with the two main objectives of monitoring the natural frequency of SS304 in defect and without defect specimens by spray coating the structure. And also monitor human health by sol-gel coated lycra denim sensor. I find these two objects not similar at all and do not know why the authors did not break the article into two instead. Additionally, this made the article very lengthy and boring. The science in the article is sound and indeed warrants publishing. However, the ideas need to be made precise and easy to follow. See other comments

The authors should discuss and explicitly explain the term “natural frequency” at first mention. Most of the article is about this, but no relative descriptions have been made

The work is overly explained. It looks like a thesis to me, not a research article. The authors should summarise the work and avoid un relevant facts and discussions—especially those common in the field.

Also, try to keep the discussion short and precise. In the same way, you should focus the paper on key results and push the rest to supplementary information

If possible, break the article into two. Probably if you want to publish the work at the same time with the same discussions, you can make twin publications with the same title but with “Part 1 natural frequency sensing” and “Part 2 human health monitoring”.

Language and grammar should be improved

Reviewer 2 Report

Review ofSmart graphene nanoplatelet strain sensor for natural frequency sensing of stainless steel (SS304) and human health monitoring" by Debadatta Sethy et al.

The authors used graphene nanoplatelet (GNPs) doped poly (methyl methacrylate) (PMMA) as a strain sensor to monitor the natural frequency of stainless steel (SS304) with and without defect, and GNPs-coated lycra denim textiles by sol-gel solutions as a human health sensor. Although the research topic seems interesting, there is a need to revise the manuscript comprehensively. I have some comments for improving the quality of this research paper for publication in the materials journal. You will find in the following my detailed comments:

  1. There are various reports on the application of graphene as a pressure sensor. What makes this work different from the previously published research? What is the novelty of this research?
  2. Line 136-137: “It tells about the presence of metha-acrylate which has three intensity peaks, one at 13.6Ëš another at 23Ëš and at 42.6Ëš which says about the amorphous nature of PMMA. The sensor solution (GNPs/PMMA/THF) also shows the intensity peak because of the presence of PMMA, GNPs and THF.” If the authors believe three peaks at 13.6Ëš, 23Ëš and at 42.6Ëš are regarding the amorphous nature of PMMA, why does the red curve in Figure 2d not show these peaks? In the blue curve, which peak indicates GNPs?
  3. Please replace figure 4 with a higher resolution figure. The inset figures are not clear.
  4. Figures 5, 6, 7: the figures' legends are not readable. Please increase the font size and resolution.
  5. Line 210-224: in these lines, the authors discuss the results of change in electrical resistance. When the authors are talking about the results, please name the panel discussing the result.
  6. As can be seen from the results, this manuscript has two major sections. The first section is about applying GNP/PMMA as a nanocomposite strain sensor and the second section is about the application of GNPs-coated lycra denim textiles by sol-gel solutions as a human health sensor. What is the connection between these two sections? Why didn’t the authors put the results in two different manuscripts?
  7. The Abstract and Conclusion must be written in an understandable form without referring to the results inside the manuscript. When the authors use positions 1, 2, etc., it is not easy to follow the Abstract and Conclusion. The authors, please use a more general format for explaining the results in the Abstract and Conclusion.
  8. All abbreviations used in the manuscript (chemical compounds, technique, device, etc.) must be defined when the authors use them for the first time in the manuscript, i.e., GF, FFT, FEM, etc.
  9. The introduction section must be improved. I recommend the authors read this article and use it in the "Introduction": https://doi.org/10.1016/j.jallcom.2019.07.187; https://doi.org/10.1016/j.jallcom.2021.163176

Best wishes,

Reviewer

Reviewer 3 Report

  1. what were the conditions during spray coating?
  2.  Have the authors checked the adhesion of coated films?
  3. What were the parameters during spray coating and spin coating?
  4. Is there any temperature of treatment was carried out after spray coating.
  5. Have the authors studied the repeatability of the fabricated sensors.
  6. Authors have to improve the quality of their results related to all the figures.

Round 2

Reviewer 1 Report

Acceptable amendments made as per comments. Can be considered for acceptance

Reviewer 2 Report

Review ofSmart graphene nanoplatelet strain sensor for natural frequency sensing of stainless steel (SS304) and human health monitoring" by Debadatta Sethy et al.

The authors used graphene nanoplatelet (GNPs) doped poly (methyl methacrylate) (PMMA) as a strain sensor to monitor the natural frequency of stainless steel (SS304) with and without defect, and GNPs-coated lycra denim textiles by sol-gel solutions as a human health sensor. In my opinion, the manuscript has been modified well, and all my previous concerns have been well responded to.

Best wishes,

Reviewer